# UTAH: Using Telemedicine to improve early medical Abortion at Home: a protocol for a randomised controlled trial comparing face-to-face with telephone consultations for women seeking early medical abortion

John Joseph Reynolds-Wright ®,[1,2] John Norrie,[3] Sharon Tracey Cameron ®[1,2,4]

For numbered affiliations see end of article.

**Correspondence to**
Dr John Joseph Reynolds-Wright; jjrw@doctors.org.uk

## ABSTRACT

**Introduction** Early medical abortion (EMA) is a two-stage process of terminating pregnancy using oral mifepristone (a progesterone-receptor antagonist) followed usually 1–2 days later by sublingual, vaginal or buccal misoprostol (a prostaglandin analogue). There are no published randomised controlled trials (RCTs) on the use of telemedicine for EMA. Our proposed research will determine if telephone consultations for EMA (the most common method of abortion in the UK) is non-inferior to standard face-to-face consultations with regard to the efficacy of EMA.

**Methods and analysis** This study will be conducted as an RCT. The recruitment target is 1222 participants. The primary outcome is success of EMA (complete abortion rate). This will be determined based on a negative low-sensitivity urine pregnancy test result (2 weeks after misoprostol use) and absence of surgical intervention or diagnosis of ongoing pregnancy (within 6 weeks of misoprostol).

Secondary outcomes include total time spent at a clinic appointment to receive EMA, self-reported preparedness for EMA, level of satisfaction with consultation and effective contraception uptake compared with when women attend for a face-to-face consultation.

The main analysis will be a modified intention-to-treat analysis. This will include all randomised women (with a viable pregnancy) using EMA and follow-up for the main outcome. The study initiated on 13 January 2020 and is anticipated to finish in late 2021.

**Ethics and dissemination** Ethical approval was given by the South East Scotland NHS Research Ethics Committee, reference: 19/SS/0111. Results will be published in peer-reviewed journals, presented at clinical and academic meetings, and shared with participants via the clinic website.

**Trial registration number** NCT04139382.

### Strengths and limitations of this study

► This is the first study devised as a randomised controlled trial to evaluate the safety and efficacy of telemedicine in early medical abortion.
► The study design is aligned with the CONSORT (Consolidated Standards of Reporting Trials), SPIRIT (Standard Protocol Items: Recommendations for Interventional Trials) and Medical Abortion Reporting of Efficacy (MARE) recommendations in order to maximise the rigour and quality of the trial.
► The large sample size will allow a statistically and clinically meaningful analysis of the results.
► Due to the legal requirement for administration of mifepristone in a clinical facility, participants in the telemedicine arm of the study will attend clinic to receive medications.
► The primary outcome relies on participant report of low-sensitivity urine pregnancy test result.

## INTRODUCTION

We plan a trial comparing telephone consultations for women (see box 1) requesting early medical abortion (EMA—under 10 weeks pregnant) to regular face-to-face consultations. In Scotland in 2018, 7 out of 10 women having an abortion chose EMA.[1] In many settings, including Scotland, the clinic visit for a consultation to discuss a request for EMA is lengthy. Women can struggle with time off work or childcare for daytime appointments. There is evidence from observational studies that telephone consultations for EMA may be a safe and acceptable alternative.[2–6] In our study, women seeking EMA will be randomised to face-to-face (standard care) or a planned telephone consultation (in advance of the clinic visit). We will determine the success of EMA in both groups, women's satisfaction with the consultation and possible advantages and disadvantages of the telephone consultation. If the study

shows that success of EMA is maintained with a telephone consultation and that this model is acceptable to women, then this may change EMA provision throughout Scotland and other countries.

**Box 1  Language regarding gender**

Within this protocol we use the terms woman and women's health. However, it is important to acknowledge that there are people other than women for whom it is necessary to access women's health and reproductive services in order to maintain their gynaecological health and reproductive well-being. Gynaecological, sexual and reproductive health services must be appropriate, inclusive and sensitive to the needs of those individuals whose gender identity does not align with the sex they were assigned at birth.

Adapted from the Royal College of Obstetricians and Gynaecologists, Faculty of Sexual and Reproductive Healthcare & British Society of Abortion Care Providers *Abortion care in COVID-19* guidance.[22]

## Background

Abortion care is common, with approximately one in three women experiencing abortion in their lifetime worldwide.[7] Each year approximately 200 000 abortions are performed in the UK and around 13 000 of these are in Scotland.[8] Ninety-nine per cent of abortions are delivered by the National Health Service (NHS) in Scotland,[1] compared with England and Wales, where 70% are delivered outside the NHS by the independent sector.[8] Furthermore, Scotland has higher uptake of medical methods of abortion compared with England and Wales. In Scotland in 2017, 80% of all abortions were conducted in early pregnancy (under 10 weeks) and over 90% of these were EMA.[1] The WHO recommend that women can reliably self-manage much of EMA with support from a clinician.[9]

In Scotland, women who choose an EMA, typically make a single visit to a clinic for a consultation and for assessment of gestation, receipt of mifepristone (to be administered in clinic, as per UK legal requirements) and misoprostol (to self-administer at home), receipt of contraception and instructions on how to self-assess the success of the abortion (using a self-performed urinary pregnancy test).[10–12] This clinic visit can last 2–3 hours; much of which may be time spent in the waiting room. Moreover, a significant proportion of consultation time is standard history taking and information giving and could be delivered via the telephone, an app or video call rather than face-to-face.

Telephone consultations could add flexibility for women (eg, consultation in the evening), reduce footfall in clinics (shorter time spent in clinic) and allow for more flexible staff working (office working, evening working, etc). There is observational evidence from other countries where abortion is legal to support use of telemedicine (including telephone consultations) for assessment of EMA.[2–6] It is also possible that the consultation in advance of a clinic visit (for confirmation of decision,

ultrasound and to collect medications) could mean that the subsequent clinical encounter is shorter, with possible efficiencies for the service, such as more effective use of medical staffing. It may also be easier to discuss and provide ongoing contraception at this encounter as women will have had time to digest the information about EMA provided at the telephone consultation. There is some observational evidence that telephone counselling may be associated with higher uptake of post abortion contraception.[13] This could translate into fewer subsequent unintended pregnancies for women. Around 2400 abortions take place in NHS Lothian annually[1] and most women (80%) attend a community abortion service at Chalmers Centre. In 2018, over 70% of abortions in this service were EMA.[14]

We wish to determine if telephone consultations for assessment of women who are potentially eligible for EMA are non-inferior to face-to-face consultations (in terms of successful outcome of EMA). We designed a study of a telephone consultation assessment service via a randomised controlled trial (RCT). This has not been conducted before. This will be conducted within the framework of the 1967 Abortion Act.[15] This RCT will provide robust data to support future service development nationally. Telephone consultations may make abortion services more accessible for women (especially those with work or child care commitments and vulnerable women). There is the possibility that services will become more efficient and so be able to provide cover for 'remote' services at other sites or health boards. Women could have an ultrasound for gestational dating and any other tests locally, but with consultations delivered by telephone. The aims of this study are in line with the current Scottish Government policy on realistic medicine and on greater use of telemedicine services.[16] Our Patient and Public Involvement group have helped develop this protocol and will continue to be involved throughout.

## Rationale for study

There are no published RCTs on use of telemedicine for EMA. The existing evidence base is observational and exists outside of the NHS healthcare framework and outside of the UK medicolegal framework. There are only five studies that report outcomes of EMA that have been conducted in settings where abortion is legal (USA, Canada, Australia) with much heterogeneity.[2–6] Our proposed research has the potential to confirm that telemedicine for EMA (the most common method of abortion in Scotland) is non-inferior to standard face-to-face consultation with regard to efficacy.

There are no common outcome sets for abortion care research. An initiative to develop this is currently underway but is not scheduled to be complete until late 2021.[17] In the absence of a common outcome set, we selected efficacy of EMA as the primary outcome as recommended by the Medical Abortion Reporting of Efficacy (MARE) guidance.[18] We hypothesise that inferior consultations could have an impact on women's ability to

self-manage EMA and so wish to determine whether telemedicine consultations are inferior to face-to-face consultations with regard to efficacy. This RCT will gather robust data regarding success of EMA, duration of consultations, women's satisfaction with the consultation and uptake of effective contraception post abortion. These outcomes were identified from previous studies and developed in partnership with patients and public. The questionnaires used to collect this information were developed by the research team and reviewed and amended by our patient and public involvement group (Abortion Rights Edinburgh). The questionnaires were piloted with a group of patients and refined prior to the formal launch of the study.

These findings can be used to inform service development and abortion care strategy at a national level in Scotland and elsewhere, potentially impacting on the delivery of abortion care in many legal and restricted settings.

The primary research question is 'Is a telemedicine consultation for EMA non-inferior to a face-to-face consultation?' The secondary research question is 'How do the consultations compare with regard to patient satisfaction, time taken, and uptake of effective methods of contraception?'

## METHODS AND ANALYSIS
### Study design
This study will be conducted as an RCT to compare telemedicine, specifically by telephone, with face-to-face consultations for women considering EMA at home.

### Primary objective
The primary objective of this study is to determine if EMA conducted following a telephone consultation is as effective (complete abortion rate) as following face-to-face consultation.

### Secondary objectives
This study also aimed to determine if a telephone consultation for EMA is associated with less total time spent at a clinic appointment to receive EMA, preparedness for EMA, level of satisfaction with consultation, rate of unscheduled contact with care and effective contraception uptake compared with when women attend for a face-to-face consultation.

### Primary endpoint
Success of EMA as defined by complete abortion rate without surgical intervention. This will be determined based on self-reported negative low-sensitivity urine pregnancy test result (2 weeks after misoprostol). The clinical database will be reviewed at 6 weeks post misoprostol to confirm final outcome of pregnancy and any admission or surgical intervention.

### Secondary endpoints
► Women's reported 'preparedness' for EMA as assessed by pre-abortion questionnaire, when they collect their pack of medications.
► Satisfaction with consultation type as assessed by post-consultation questionnaire, conducted by telephone at 2 weeks.
► Uptake of effective contraception after EMA as assessed by case note review.
► Total time spent in clinic (both telephone and face-to-face groups) and time taken for telephone consultation.
► Unscheduled contact with abortion service or hospital within 6 weeks of EMA for concern related to EMA.

### Study population
A total of 1222 participants randomised to receive telephone consultation (n=611) or face-to-face (n=611).

The success of EMA (primary outcome – complete abortion without surgical intervention) is assumed as 97%, based on review of success rates in our regional database, as success rates in the literature are reported variably (usually between 95% and 99%). The recruitment target has been calculated using a binary outcome non-inferiority calculator with 90% power, one-sided 5% level of significance, 3% non-inferiority limit, 1:1 allocation and 10% compensation for loss to follow-up.[19] This will give us an adequately powered sample that will show statistical significance in efficacy findings.

The NHS Lothian abortion service cares for approximately 2400 women each year and of those 70% would be eligible to participate in the study. Over 18 months we should achieve adequate recruitment even if 50% of potential participants decline to participate and so should be feasible to complete within the projected timeframe.

### Inclusion criteria
► Self-reported last menstrual period less than 10 weeks on the day of appointment.
► Self-referral to Lothian Abortion Referral Service (LARS).
► Aged 16 or older at the time of the procedure.
► Preference for EMA.
► Ability to give informed consent.

### Exclusion criteria
► Requires interpreter.
► Patient preference for surgical method of abortion.

### Identifying participants
The administrative staff of LARS will collect the routine demographic information, basic obstetric history and contact details from women who self-refer for abortion (by telephone) and give them the next available date for the clinic so that participants in both study arms will receive an ultrasound scan, blood tests and sexual health screening as per usual care.

For women who meet the inclusion criteria, administrative staff will then read a short script about the study. If

women express interest in participating, then permission will be sought for the research doctor or nurse to contact them by telephone at a convenient time to woman to discuss study participation. Interested women will also be directed to the clinic website where they can read the Participant Information Sheet (PIS) and consent form in advance of the call from the research doctor or nurse.

### Consent

Consent will be obtained from participants by the research doctor or nurse verbally over the telephone using a standard form. The participant will then be randomised to receive either a telephone consultation or a face-to-face consultation. When participants arrive in clinic, they will be asked to sign an affirmation that they continue to consent in the project.

The Participant Information Sheet and Consent Form are available as online supplemental appendix 1.

Randomisation lists will be generated by the Edinburgh Clinical Trials Unit (ECTU) and randomisation is performed by the research staff using REDCap (Research Electronic Data Capture) software hosted at the University of Edinburgh.[20 21]

### Withdrawal of participants

Participants are free to withdraw from the study at any point or a participant can be withdrawn by the investigator should they no longer meet the inclusion/exclusion criteria for the study. If withdrawal occurs, the primary reason for withdrawal will be documented in the participant's case report form, if possible. The participant will have the option of withdrawal from all aspects of the trial but continued use of data collected up to that point. To safeguard rights, the minimum personally identifiable information possible will be collected.

### Study assessments

Study assessments are detailed in table 1. There is no long-term follow-up. Participants are followed up at 2 weeks post abortion only. Questionnaire 2 will be primarily conducted by telephone; however, if women are not able to answer the telephone, we will offer the option to receive the questionnaire via email or post to maximise response rate. Some study outcomes will be retrieved from routinely collected clinical data and not included in this table.

Questionnaires 1 and 2 are available as online supplemental appendices 2 and 3.

### Data collection

Baseline characteristics: Demographics, reproductive history and gestational age (based on ultrasound) will be collected on all participants.

Consultation time: Duration of telephone consultation (minutes) and duration of face-to-face clinic consultation (minutes), total time spent in clinic on day of attendance for assessment (minutes).

Participant preparedness questionnaire: At clinic on first attendance—research nurse or doctor administered questionnaire to assess how prepared they feel.

Participant acceptability questionnaire: At 2 weeks post abortion—research nurse administered telephone questionnaire using validated questions on acceptability of consultation. Alternatively, this can be self-completed online or a paper postal questionnaire (if participant is unavailable via telephone or expresses a strong preference for this mode).

Outcome of abortion: Self-reported outcome of routine low-sensitivity urine pregnancy test at 2 weeks, plus review of clinical database at 6 weeks to confirm final outcome of pregnancy.

Unscheduled contact (in person or telephone) with abortion service or hospital for concern related to EMA within 6 weeks (clinical records review at 6 weeks).

| **Table 1** Study assessments | | | | |
|---|---|---|---|---|
| **Assessment** | **When** | **Administered by** | **Description** | **Study arm** |
| Consultation duration | During telephone consultation/face-to-face consultation | Research doctor or nurse | Duration of face-to-face/telephone assessment consultation plus time spent in clinic on day of attendance. | Both arms |
| Questionnaire 1 | At the abortion clinic, following consultation prior to commencing abortion | Research doctor or nurse | A researcher-administered questionnaire identifying how prepared participant feels for EMA, how satisfied they were with consultation and plans for contraception following EMA. Demographic information will also be collected at this point. | Both arms |
| Questionnaire 2 | Over the telephone/online/by post 14–20 days following EMA | Research nurse or doctor or self | A researcher administered questionnaire to assess outcome of abortion by self-reported LSUPT outcome, satisfaction with whole abortion process and contraceptive outcome. | Both arms |

EMA, early medical abortion; LSUPT, Low-Sensitivity Urine Pregnancy Test.

## Data management

### Personal data

The following personal data will be collected as part of the research, we note that these data are already routinely collected in clinical practice as part of clinical history:

1. Name
2. Post code (in order to convert to the Scottish Index of Multiple Deprivation)
3. Weight, height, BMI
4. Previous pregnancy history
5. Physical personal data will be stored by the research team at Chalmers Centre, NHS Lothian, in the research office, behind a locked door that requires an ID badge to access and inside a locked cabinet in the room.

Study participants are assigned a numerical code to act as their identifier and is used when recording responses on paper and electronic data capture forms.

Electronic personal data will be kept on an NHS Lothian shared drive in password-protected files. Passwords will be kept by research team and a hard copy with the locked physical data.

Identifiable personal data will be stored for a maximum of 5 years. Totally deidentified data will be retained for 10 years in total.

Data will be shared with colleagues at the University of Edinburgh Clinical Trials Unit (ECTU) who will assist with database management and statistical support.

### Transfer of data

Data collected or generated by the study (including personal data) will not be transferred to any external individuals or organisations outside of the sponsoring organisations.

### Data controller

The University of Edinburgh and NHS Lothian are joint data controllers.

### Data breaches

Any data breaches will be reported to the University of Edinburgh and NHS Lothian Data Protection Officers who will onward report to the relevant authority according to the appropriate timelines if required.

## Statistics and data analysis

### Proposed analyses

Statistical analysis will be conducted in partnership with the Edinburgh Clinical Trials Unit, University of Edinburgh.

Descriptive statistics will be used to characterise participants and assess comparability of the two groups at baseline.

For the primary outcome (efficacy of EMA), the main analysis will be a modified intention-to-treat analysis. This will include all randomised women, undergoing medical abortion, with a viable pregnancy (ie, not ectopic, molar), and follow-up for the main outcome recorded within 6 weeks of the abortion treatment.

A sensitivity analysis will be performed on an intention-to-treat population consisting of all randomised women having had medical abortion with viable pregnancy. We will impute the outcome for women lost to follow-up.

Secondary outcomes will be analysed using appropriate tests depending on the normality of the data: for normally distributed data, we plan to use independent and paired t-tests; for non-normally distributed and categorical data, we plan to use a combination of Mann-Whitney, Kruskal-Wallis and $\chi^2$ testing. Results will be considered statistically significant if p value <0.05.

No interim analysis is planned.

## Patient and public involvement

We consulted Abortion Rights Edinburgh, a local abortion and women's rights activism group. They kindly provided feedback on the trial rationale, study design and study protocol prior to submission for ethical approval. They have agreed to disseminate the trial findings to their membership and via their networks.

## ETHICS AND DISSEMINATION

### Ethical approval

Ethical approval has been granted by South East Scotland NHS Research Ethics Committee on 28 October 2019, reference: 19/SS/0111.

### Dissemination plan

Results will be published in peer-reviewed journals and as presentations at national and international meetings. All data will be reported in full. Participants will be able to access a summary of the trial results via the clinic website. Abortion Rights Edinburgh will disseminate to their membership and networks. The findings are likely to influence national and international guidance on best practice provision of abortion care.

### Study status

The study opened to recruitment on 13 January 2020 and is temporarily paused due to service, legal and clinical guidance changes during COVID-19, meaning that all patients are currently receiving telemedicine care.[22] The status of telemedicine care is under legal review in Scotland (and England and Wales) and the outcome of this is expected later in 2021 and will determine whether recruitment can recommence.

### Administrative details

UTAH (Using Telemedicine to improve early medical Abortion at Home) was registered with clinicaltrials.gov on 25 October 2019 (unique identifier: NCT04139382).

UTAH is jointly sponsored by the University of Edinburgh (UK) and NHS Lothian (UK) via the ACCORD partnership and assigned the identifier AC19076. Protocol version: 1.0; Date: 18 September 2019.

The sponsor reviewed the study design and gave research and development approval to the trial. They are not involved in the collection, management, analysis or

interpretation of the data, nor will they be involved in any report writing.

The research team are John Reynolds-Wright (Clinical Research Fellow), Anne Johnstone (Clinical Research Nurse), Karen McCabe (Clinical Research Midwife), Claire Nicol (Lead Nurse, Abortion Service) and Sharon Cameron (Principle and Chief Investigator).

**Author affiliations**
[1]The Queen's Medical Research Institute, The University of Edinburgh MRC Centre for Reproductive Health, Edinburgh, UK
[2]Chalmers Centre for Sexual and Reproductive Health, NHS Lothian, Edinburgh, UK
[3]Edinburgh Clinical Trials Unit, University of Edinburgh, Edinburgh, UK
[4]Obstetrics and Gynaecology, University of Edinburgh, Edinburgh, UK

**Contributors** JJR-W and STC equally contributed to the design of the protocol. JN contributed to the statistical analysis and sampling sections of the protocol. All authors reviewed the final manuscript for submission.

**Funding** This work was supported by the Edinburgh Family Planning Trust grant number EFPT/2019/UTAH. The project was conducted from the MRC Centre for Reproductive Health, supported by the Medical Research Council (grant MR/N022556/1).

**Competing interests** Professor JN was Deputy Chair of the National Institute of Health Research (NIHR) Health Technology Assessment (HTA) General Funding Committee (2016-2019) and is currently Chair of the Medical Research Council (MRC)/NIHR Efficacy and Mechanisms Evaluation (EME) Funding Committee.

**Patient consent for publication** Not required.

**Provenance and peer review** Not commissioned; externally peer reviewed.

**ORCID iDs**
John Joseph Reynolds-Wright http://orcid.org/0000-0001-6597-1666
Sharon Tracey Cameron http://orcid.org/0000-0002-1168-2276

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
