## [Reviewer comments · BMJ Open]

ARTICLE DETAILS

TITLE (PROVISIONAL)	UTAH: Using Telemedicine to improve early medical Abortion at Home – a protocol for a randomised controlled trial comparing face-to-face with telephone consultations for women seeking early medical abortion.
AUTHORS	Reynolds-Wright, John; Norrie, John; Cameron, Sharon

VERSION 1 – REVIEW

REVIEWER	Norman, Wendy V. University of British Columbia, Dept of Family Practice
REVIEW RETURNED	06-Dec-2020

GENERAL COMMENTS	Thank you for the opportunity to review this interesting and well articulated manuscript. This review was primarily conducted by my post-doctoral Fellow, Madeleine Ennis, PhD, under my supervision and we discussed together and accepted our final comments to you below. best wishes, Wendy V. Norman Summary: The authors aimed to describe the protocol for a RCT comparing in-person vs telephone consultations for individuals seeking medical abortion care at less than 10 weeks gestation. This RCT began in early 2020, and is paused due to the Covid-19 pandemic. The publication of this protocol increases study transparency, could facilitate collaboration, and allow researchers and clinicians to stay up-to-date with current research activity and issues in modern abortion care. After a Pubmed search, I concluded that this is original research, with unique objectives. This publication would be extremely relevant to BMJ readers (many of whom would be abortion providers), keeping them up-to-date on abortion research that has the potential to change clinical practice. This being said, there are some substantial changes that should occur in the description of the protocol prior to publication. Firstly, the language in the manuscript should be altered to reflect biological sex, and not gender (e.g. refer to females instead of women when making general statements). This language recognizes diversity and is more inclusive. While the study design seems appropriate to answer the research questions, I would recommend a more in-depth rationale of the selected outcomes and assessments, with additional references. For example, providing references for, or a more in-depth description of, the development of the questionnaires. As well, the authors need to provide a description of limitations, such as the potential for bias (selection bias, reporting bias, etc). While some limitations and strengths are listed within the abstract, they are not directly listed in the manuscript, where they belong. Removing them from the
---

	abstract would also allow for description of the secondary outcomes, and assist in creating a more balanced and complete abstract. Additionally, while they have provided a knowledge translation plan, there is limited discussion of expected outcomes and impact. The authors have provided excellent details the sample size calculation, however, elaborating on the description of the post data collection analysis would allow readers to better assess the degree of scientific rigor. While the study appears well designed, we are missing key details and rationales to have full understanding of this RCT. Lastly, there is missing punctuation and a number of typos that should be corrected. Comments: Overall:  1. Be consistent with language around telemedicine, potentially providing definitions for each term used: telemedicine vs telephone abortion vs telephone consultation, etc. 2. References are out of order, reference 7 should come before reference 8. 3. Are there other potential outcomes that you could have chosen to assess the non-inferiority of telemedicine compared to in person clinic appointments? Please describe this decision. Abstract:  1. Page 4, line 29: Currently saying “2 weeks days,” should be “2 weeks” 2. Page 5, line 6-7: Please reword, language is confusing. 3. I recommend shortening the description of sample size calculations and removing the point form list of strengths and limitations, and elaborating on some of your secondary objectives. This information is more relevant for the audience of BMJ. Background:  1. Please provide references for your first two sentences in the background. Are 1 in 3 women accessing abortion care in Scotland, UK or globally? 2. Page 6, line 38: change language from wish. Suggestions: “women who access an EMA” or “women who choose an EMA.” 3. Page 6, line 43: Remove “waiting to be seen,” redundant 4. This sentence should be reworded, it is confusing and lengthy: “Additionally, if women are better informed about EMA in advance of the clinic, the clinic consultation may be better utilised to answer outstanding questions that the woman may have or discuss and provide ongoing contraception.” 5. Page 7, lines 16-22: reduce sentence. Study design:  1. Are you reviewing the clinical database at 1 month or 6 weeks post abortion? This differs between abstract and study design section. Study population:  1. Page 8, line 5: Why is the success of EMA assumed at 97%. Needs reference and explanation. 2. Page 9, lines 6-8: Please justify the use of 50%. Do you have a reference for this? 3. For inclusion and exclusion criteria, provide details on potential participants who ended up experiencing pathological (eg ectopic) pregnancies. Method of abortion not solely based on patient preference. Identifying participants:
--	---

	1. Page 9, line 27: “Usual details” is too vague and not scientific language. Would recommend changing to “medical history” or similar. Withdrawal of participants: 1. Please provide an explanation as to why the investigator would ask a participant to withdraw from the study. Likely due to them no longer meeting the inclusion/exclusion criteria? Important to explain, to give better understanding of potential bias. Study assessments: 1. Please provide justification for why the Questionnaire 2 can be conducted over the telephone, online, or by post. Consistency in questionnaire administration is important, and there are participants may provide different answers when speaking to a person vs writing in a questionnaire. 2. Why are the assessment details for some secondary objectives (eg contraception uptake) not included in Table 1? Are these only detailing assessments with participant interaction? If so, this should be clarified below the table. Data collection: 1. What analysis will be run on the unscheduled contact with abortion service or hospital? Which secondary objective does this relate to? Is documenting unexpected complications one of your secondary objectives, unlisted? Data Management: 1. You state in Withdrawal of Participants that “minimum personally identifiable information possible” will be collected. Please justify the collection of identifiers such as postal code. Do you use Alpha-numeric codes, with personal identifiers kept separate from the data? Statistics and data analysis: 1. Page 12, lines 26-28: Please be more specific about analyses to be conducted after analyzing for normality. This specification speaks to a well thought out data analysis plan. Study Status: 1. Are there plans on when to reinitiate the study? Will data collected prior to Covid-19 still be analyzed with data collected after? Covid-19 has changed the abortion telemedicine landscape.
--	--

REVIEWER	Mazza, Danielle Monash University, General Practice
REVIEW RETURNED	16-Dec-2020

GENERAL COMMENTS	Thank you for the opportunity to review this clearly written protocol paper. It systematically addresses all the requirements of a protocol for a RCT. I have three comments/questions: 1. it would be good to have access in a supplementary file to the patient preparedness and acceptability questionnaires. Could the authors please provide a reference/s for the study/ies which validated these questionnaires? 2. Could the authors explain why in the telemedicine arm there is still a requirement for patients to attend the clinic to pick up their medication. Could this not be posted to them or could they not pick it up directly from a pharmacy? If not why not? 2. The authors advise that the study has been paused due to COVID. Could they explain why as women will still be presenting for abortions during this time
---

VERSION 1 – AUTHOR RESPONSE

Reviewer: 1

Dr. Wendy V. Norman, University of British Columbia

Comments to the Author:

Thank you for the opportunity to review this interesting and well articulated manuscript. This review was primarily conducted by my post-doctoral Fellow, Madeleine Ennis, PhD, under my supervision and we discussed together and accepted our final comments to you below. best wishes, Wendy V. Norman

Summary:

The authors aimed to describe the protocol for a RCT comparing in-person vs telephone consultations for individuals seeking medical abortion care at less than 10 weeks gestation. This RCT began in early 2020, and is paused due to the Covid-19 pandemic. The publication of this protocol increases study transparency, could facilitate collaboration, and allow researchers and clinicians to stay up-to-date with current research activity and issues in modern abortion care. After a Pubmed search, I concluded that this is original research, with unique objectives. This publication would be extremely relevant to BMJ readers (many of whom would be abortion providers), keeping them up-to-date on abortion research that has the potential to change clinical practice.

Thank you

This being said, there are some substantial changes that should occur in the description of the protocol prior to publication. Firstly, the language in the manuscript should be altered to reflect biological sex, and not gender (e.g. refer to females instead of women when making general statements). This language recognizes diversity and is more inclusive.

Thank you for highlighting this. There are of course merits in phrasing in relation to biological sex, but even this is not absolute and labelling as 'female' can be unacceptable to trans men and non-binary people.

We also feel that it remains important to centre cis women in abortion care and research, as their sexual and reproductive health remains marginalised and stigmatised by our heteropatriarchal society.

BMJ Open does not appear to have a policy in relation to this and so we have added in the language used by the Royal College of Obstetricians and Gynaecologists (RCOG - UK) and Faculty of Sexual and Reproductive Healthcare (FSRH - UK) as 'Box 1' below. We further acknowledge that this language is not perfect but wish to adopt it here as it is in line with the UK organisations responsible for abortion care.

Box 1: Language regarding gender

Within this document we use the terms woman and women's health. However, it is important to acknowledge that it is not only people who identify as women for whom it is necessary to access women's health and reproductive services in order to maintain their gynaecological health and reproductive wellbeing. Gynaecological and obstetric services and delivery of care must therefore be appropriate, inclusive and sensitive to the needs of those individuals whose gender identity does not align with the sex they were assigned at birth.

While the study design seems appropriate to answer the research questions, I would recommend a more in-depth rationale of the selected outcomes and assessments, with additional references. For example, providing references for, or a more in-depth description of, the development of the questionnaires.

We have elaborated on this in the 'rationale for study section' – we note an error in the text that stated 'validated questionnaires' which should not have been included as the questionnaires have not been validated.

As well, the authors need to provide a description of limitations, such as the potential for bias (selection bias, reporting bias, etc). While some limitations and strengths are listed within the abstract, they are not directly listed in the manuscript, where they belong. Removing them from the abstract would also allow for description of the secondary outcomes, and assist in creating a more balanced and complete abstract.

The bullet-point list below the abstract for strengths and limitations is required by the BMJ Open format and so must remain.

Additionally, while they have provided a knowledge translation plan, there is limited discussion of expected outcomes and impact.

We have added a line to the rationale and dissemination sections to mention the potential impact of the study

The authors have provided excellent details the sample size calculation, however, elaborating on the description of the post data collection analysis would allow readers to better assess the degree of scientific rigor.

We have added further detail to this section as requested.

While the study appears well designed, we are missing key details and rationales to have full understanding of this RCT. Lastly, there is missing punctuation and a number of typos that should be corrected.

We have proof-read the manuscript and have corrected typos and punctuation.

Comments:

Overall:

1. Be consistent with language around telemedicine, potentially providing definitions for each term used: telemedicine vs telephone abortion vs telephone consultation, etc.

Thanks for highlighting this – we have harmonised the terminology to be ‘telemedicine’ and then in study design elaborate that this will be specifically by telephone for the current study.

2. References are out of order, reference 7 should come before reference 8.

Thank you – we have corrected this and the reference list.

3. Are there other potential outcomes that you could have chosen to assess the non-inferiority of telemedicine compared to in person clinic appointments? Please describe this decision.

We have elaborated on this in the ‘rationale’ section.

Abstract:

1. Page 4, line 29: Currently saying “2 weeks days,” should be “2 weeks”

Thank you, now corrected

2. Page 5, line 6-7: Please reword, language is confusing.

We believe you are referring to this point (the line numbers on the pdf seem to span more than 1 point):

- Due to the medico-legal framework surrounding abortion, the telemedicine arm of the study still needs to attend in person to receive medications.

And we have modified to the following:

- Due to the legal requirement administration of mifepristone in in a clinical facility, participants in the telemedicine arm of the study will attend clinic to receive medications.

Please let us now if this was not the correct point to modify.

3. I recommend shortening the description of sample size calculations and removing the point form list of strengths and limitations, and elaborating on some of your secondary objectives. This information is more relevant for the audience of BMJ.

We have shortened the sample size calculation and added information on the secondary outcomes, however the 'Strengths and Limitations' section is part of the BMJ Open format and must remain in this place.

Background:

1. Please provide references for your first two sentences in the background. Are 1 in 3 women accessing abortion care in Scotland, UK or globally?

We have added references and clarified this statement further

2. Page 6, line 38: change language from wish. Suggestions: "women who access an EMA" or "women who choose an EMA."

We have altered this to 'choose'

3. Page 6, line 43: Remove "waiting to be seen," redundant

This has been removed

4. This sentence should be reworded, it is confusing and lengthy: "Additionally, if women are better informed about EMA in advance of the clinic, the clinic consultation may be better utilised to answer outstanding questions that the woman may have or discuss and provide ongoing contraception."

Thank you – we have restructured this sentence: It may also be easier to discuss and provide ongoing contraception at this encounter as women will have had time to digest the information about EMA provided at the telephone consultation.

5. Page 7, lines 16-22: reduce sentence.

We have broken this into several shorter sentences

Study design:

1. Are you reviewing the clinical database at 1 month or 6 weeks post abortion? This differs between abstract and study design section.

The database is checked at 6 weeks to confirm final pregnancy outcome and to detect admission or surgical intervention. This has been clarified within the text in the study design section.

Study population:

1. Page 8, line 5: Why is the success of EMA assumed at 97%. Needs reference and explanation.

We defined success of medical abortion as complete abortion without surgical intervention. In the literature the success rate for medical abortion is quoted between 95 and 99% in various studies that do not fully capture the gestational age group in this study (up to 9 weeks + 6days) and inconsistently report success in the way we have defined. As such, we reviewed our local clinical database and determined that the success rate in local practice is approximately 97%, but we do not have a published reference for this. We have added text to this section to clarify.

2. Page 9, lines 6-8: Please justify the use of 50%. Do you have a reference for this?

This statement was related to the feasibility of the study within the projected timeframe and meant to indicate that even if half of eligible women declined we would still complete recruitment on time. We have added to this line to clarify. There is no relevant reference.

3. For inclusion and exclusion criteria, provide details on potential participants who ended up experiencing pathological (eg ectopic) pregnancies. Method of abortion not solely based on patient preference.

The inclusion/exclusion criteria are for receiving the telephone consultation/participating in study rather than eligibility for EMA. If following randomisation and then consult, participant changes mind/clinically requires in-patient/surgical or ectopic is detected on scan that is then captured as an outcome but would be detected after inclusion in the study (listed under secondary endpoints).

Identifying participants:

1. Page 9, line 27: "Usual details" is too vague and not scientific language. Would recommend changing to "medical history" or similar.

We have altered this to state ...administrative staff of LARS will collect the routine demographic information, basic obstetric history and contact details from women who self- refer for abortion...

Withdrawal of participants:

1. Please provide an explanation as to why the investigator would ask a participant to withdraw from the study. Likely due to them no longer meeting the inclusion/exclusion criteria? Important to explain, to give better understanding of potential bias.

Thank you we have now explicitly stated this in the text.

Study assessments:

1. Please provide justification for why the Questionnaire 2 can be conducted over the telephone, online, or by post. Consistency in questionnaire administration is important, and there are participants may provide different answers when speaking to a person vs writing in a questionnaire.

Thank you, we have now added the following sentence:

Questionnaire 2 will be primarily conducted by telephone, however, if women are not able to answer the telephone we will offer the option to receive the questionnaire via email or post to maximise response rate.

2. Why are the assessment details for some secondary objectives (eg contraception uptake) not included in Table 1? Are these only detailing assessments with participant interaction? If so, this should be clarified below the table.

Contraception uptake is included in table 1 (Column = description, Row = Questionnaire 2). We have added a line to the text below stating: Some study outcomes are retrieved from routinely-collected clinical data and not included in this table.

Data collection:

1. What analysis will be run on the unscheduled contact with abortion service or hospital? Which secondary objective does this relate to? Is documenting unexpected complications one of your secondary objectives, unlisted?

Under study design, 'secondary endpoints' we state Unscheduled contact with abortion service or hospital within 6 weeks of EMA for concern related to EMA. We have added 'rate of unscheduled contact with care' under the secondary objectives list.

Data Management:

1. You state in Withdrawal of Participants that "minimum personally identifiable information possible" will be collected. Please justify the collection of identifiers such as postal code. Do you use Alpha-numeric codes, with personal identifiers kept separate from the data?

We have added the following sentence: Study participants are assigned a numerical code to act as their identifier and is used when recording responses on paper and electronic data capture forms.

We have also stated that we will collect the postal code (which is not individually identifiable) to convert to Scottish Index of Multiple Deprivation – a marker of socioeconomic deprivation. Further, these are all data that are captured by the health service in routine care rather than specifically for the purpose of the study.

Statistics and data analysis:

1. Page 12, lines 26-28: Please be more specific about analyses to be conducted after analyzing for normality. This specification speaks to a well thought out data analysis plan.

We have added further detail to this section.

Study Status:

1. Are there plans on when to reinitiate the study? Will data collected prior to Covid-19 still be analyzed with data collected after? Covid-19 has changed the abortion telemedicine landscape.

The study was suspended as legislation and clinical guidance changed, strongly advocating for telemedicine care wherever possible during the covid-19 outbreak. We are awaiting decision by the Scottish Government on the continued status of telemedicine abortion care. This is expected later in 2021. Should we revert to the previous medicolegal paradigm under which the study was conceived, the study will be able to resume. Should the temporary legislation remain in place permanently, the study team will meet to determine whether or not to formally close the study.

The section now reads: The study opened to recruitment on 13th January 2020 and is temporarily paused due to service, legal and clinical guidance changes during covid-19, meaning that all patients are currently receiving telemedicine care. The status of telemedicine care is under legal review in Scotland (and England and Wales) and the outcome of this is expected later in 2021 and will determine whether recruitment can recommence.

Reviewer: 2

Prof. Danielle Mazza, Monash University

Comments to the Author:

Thank you for the opportunity to review this clearly written protocol paper. It systematically addresses all the requirements of a protocol for a RCT. I have three comments/questions:

1. it would be good to have access in a supplementary file to the patient preparedness and acceptability questionnaires. Could the authors please provide a reference/s for the study/ies which validated these questionnaires?

We stated in error that the questionnaires were validated in the original submission. The questionnaires were developed locally with PPI involvement but are not validated. The questionnaires used are included as supplementary material.

2. Could the authors explain why in the telemedicine arm there is still a requirement for patients to attend the clinic to pick up their medication. Could this not be posted to them or could they not pick it up directly from a pharmacy? If not why not?

In England, Scotland and Wales, Mifepristone must be administered in a clinical facility as a requirement of the Abortion Act 1967. This is temporarily suspended during covid and is currently under review. There is potential for this requirement to be reinstated following the review. We have added that this is a legal requirement to the introduction section.

3. The authors advise that the study has been paused due to COVID. Could they explain why as women will still be presenting for abortions during this time

Due to Covid guidance we were no longer able to randomise safely as all patients were receiving telemedicine care as standard to minimise covid exposure. We have added the following text to this section: The study opened to recruitment on 13th January 2020 and is temporarily paused due to service, legal and clinical guidance changes during covid-19, meaning that all patients are currently receiving telemedicine care. The status of telemedicine care is under legal review in Scotland (and England and Wales) and the outcome of this is expected later in 2021 and will determine whether recruitment can recommence.